# Effects of Anti-Seizure Medication on Sleep Spindles and Slow Waves in Drug-Resistant Epilepsy

**DOI:** 10.3390/brainsci12101288

**Published:** 2022-09-24

**Authors:** Jennifer K. Roebber, Penelope A. Lewis, Vincenzo Crunelli, Miguel Navarrete, Khalid Hamandi

**Affiliations:** 1Cardiff University Brain Research Imaging Centre (CUBRIC), School of Psychology, Cardiff University, Maindy Rd., Cardiff CF24 4HQ, UK; 2The Welsh Epilepsy Unit, Department of Neurology, University Hospital of Wales, Heath Park, Cardiff CF14 4XN, UK; 3Neuroscience Division, School of Bioscience, Cardiff University, Cardiff CF10 3AX, UK

**Keywords:** waveform coupling, EEG, polytherapy, slow wave, drug resistant epilepsy, anti-seizure medication, memory consolidation, anti-epilepsy drugs, sleep

## Abstract

There is a close bidirectional relationship between sleep and epilepsy. Anti-seizure medications (ASM) act to reduce seizure frequency but can also impact sleep; this remains a relatively unexplored field given the importance of sleep on seizure occurrence, memory consolidation, and quality of life. We compared the effect of poly-ASM treatment on a night of sleep compared to an unmedicated night in patients with drug-resistant epilepsy, where ASMs were withdrawn and later restored as part of their pre-surgical evaluation. Within-subject analysis between medicated and unmedicated nights showed ASMs increased spindle (11–16 Hz) power and decreased slow wave (0.1–2 Hz) amplitude. Spindles became less strongly coupled to slow waves in the ASM night compared to no-ASM night, with effects to both the phase and strength of coupling and correlated with slow wave reduction. These effects were not seen in age-matched controls from the same unit where ASMs were not changed between two nights. Overall, we found that ASM polytherapy not only changed specific sleep waveforms, but also the fine interplay of spindle/slow wave coupling. Since these sleep oscillations impact both seizure occurrence and memory consolidation, our findings provide evidence towards a decoupling impact of ASMs on sleep that should be considered in future studies of sleep and memory disruption in people with epilepsy.

## 1. Introduction

Sleep is critical for maintaining mental and physical health. However, patients with epilepsy report insomnia, less sleep and worse sleep quality, affecting their quality of life and the disease burden [1,2,3,4,5,6]. Indeed, sleep disturbances are the second largest driver of low quality of life in people with epilepsy and have a larger impact on quality of life than seizure control [1,3,6]. Thus, understanding how sleep is impaired in patients with epilepsy is an important and relatively unexplored component of the disease management.

Sleep and epileptic seizures are closely related. Seizures disrupt sleep and sleep deprivation acts as a common trigger for seizures [2,7,8]. Seizures occur frequently, and in some patients exclusively, during sleep with >90% of these arising during non-rapid eye movement sleep (NREM) sleep [9]. Two sleep-specific EEG waves characterize and dominate NREM sleep: slow waves (0.5–2 Hz) and spindles (11–16 Hz). These waveforms are generated within cortico-thalamo-cortical networks and could act as a biomarker of cortical excitability [7,10,11]. One hypothesis for why seizures are selective to NREM sleep is that neuronal synchrony across the cortex is enhanced during NREM and neural synchronization is also associated with seizure initiation and propagation [7,8,12]. Conversely, ASMs directly act to reduce this high frequency phase synchronization [11]. Seizure EEG rhythms often have phenomenological and likely mechanistic similarities with sleep features, involving thalamic, hippocampal and cortical interactions. For example, ictal slow waves resemble sleep slow wave and hippocampal ictal spikes closely resemble physiological ripples [13,14]. Additionally, an important synchronization during sleep is the coupling of slow waves and spindles, which is relevant to memory consolidation [15,16,17]. Memory consolidation circuits and the seizure rhythms may interact, as seizure onset is preceded by large amplitude slow waves and reduced spindles [13,18]. It is possible that during sleep, seizures and interictal activity may hijack existing sleep networks used for memory consolidation. As a result, they could either ‘wash out’ existing memory traces or create erroneous signals [14].

Many ASM cause drowsiness or sleepiness and some cause insomnia, typically with a dose-dependent effect [19]. Benzodiazepines, in particular, which are used to treat both epilepsy and insomnia are well known to increase sleepiness and reduce REM sleep [2,20,21,22,23]. Studies of the effects of other ASM or combined ASM on human sleep in patients with epilepsy are limited, as obtaining a drug-free baseline is constrained by seizure risk and ethical considerations [2], while studies in healthy controls or patients receiving add-on ASM show variable and contrasting results [2,20]. This further differs from real-world patient outcomes, as the existing studies on healthy subjects test ASM as a monotherapy over only few nights, which is heavily contrasted against chronic polytherapy prescribed to patients [2,19,20]. Though these investigations have provided critical information about the action of individual ASM on sleep, the real-world effects of ASMs on sleep EEG features have been less frequently studied. Notwithstanding, the opportunity for such studies exists through epilepsy monitoring units where ASM are reduced or stopped during continuous EEG over several days in patients undergoing pre-surgical evaluation. This study was therefore conceived to explore modifications on sleep in patients with drug-resistant epilepsy in this setting, using existing datasets from patients who had undergone pre-surgical evaluation.

We retrospectively studied and compared sleep and EEG data from two cohorts of patients from the same epilepsy monitoring unit (EMU) who had undergone inpatient video-EEG monitoring across two nights. Group 1 (hereafter called the Epilepsy group) had a diagnosis of drug-resistant focal epilepsy and were undergoing evaluation for epilepsy surgery. The Epilepsy group has ASMs tapered during admission in order to increase the likelihood of seizure occurrence during the admission. and returned to full dose the evening prior to discharge. The tapering of ASMs was individualized for each patient according to seizure type and occurrence. Here, we selected only patients where medications had been tapered to zero during admission, before returning to full dose prior to discharge, to compare effects of no ASM to full ASM nights (Figure 1A). Group 2 were patients admitted for diagnostic clarification, we selected only those whose discharge diagnosis was of dissociative seizures (DS), also known as Non-Epileptic Attack Disorder (NEAD), based on the semiology of video recorded habitual attacks, and no abnormality on ictal or interictal EEG. In total, 86% of group 2 were also taking ASMs due to previous suspected epilepsy diagnosis. These acted as a control group (hereafter called the Control group) to examine any between-night changes in sleep parameters in the hospital environment.

## 2. Materials and Methods

### 2.1. Participants

We analyzed retrospective scalp EEG recordings from 29 patients in the EMU, University Hospital of Wales, Cardiff, which is a tertiary epilepsy center for Wales. All patients had a diagnosis of either drug-resistant focal epilepsy or a discharge diagnosis of DS and had undergone continuous video and scalp EEG monitoring for at least two sequential nights. Diagnoses were confirmed by a clinician based on EEG records and clinical notes (Appendix A). Patients with suspected obstructive sleep apnea, based on known previous history or overnight observations (snoring with apneic episodes), were excluded from the study.

Both groups had sleep recorded in the same clinical environment in the EMU. The unit is located off of the main ward, and has no nighttime admissions or rounds, and lights are off. Patients are under continuous observation by nursing staff. As part of the unit’s standard protocol patients undergoing pre-surgical evaluation have ASM individually tapered over a five-day admission (Monday to Friday) to increase the likelihood of seizure occurrence. Patients were selected for this study based on complete withdrawal of all ASMs by day 3, as per protocol, when no seizures had occurred up to that point. All ASM were re-instated on night 4 for all patients, ready for discharge home on day 5 (Friday). Patients in the Epilepsy group were taking between 1 and 4 (average of 2.3) of the following ASM at the time of the study: Clobazam (CLB), Clonazepam (CLN), Sodium Valproate (VPA), Topiramate (TPM), Lamotrigine (LTG), Levetiracetam (LEV), Zonisamide (ZNM), Gabapentin (GBP), Pregabalin (PGB), Perampanel (PMP), Lacosamide (LCM), Eslicarbazepine (ESL). Ten patients had a diagnosis of temporal lobe epilepsy, 3 frontal, 1 occipital, and 1 parietal.

The Control group had at least two nights of EEG monitoring for diagnostic clarification of epilepsy versus DS/NEAD and had no change in their medication during monitoring (Appendix A). Only patients with a discharge diagnosis of DS were included to enable a between group comparison of patients with epilepsy to those without epilepsy, as well as the within group between night comparison. EEGs and case records were reviewed, and patients were only included in the control group if there was a video-EEG recording of habitual events consistent with DS from this admission, no interictal EEG abnormalities, and no suspicion of previous epileptic seizures that might suggest a dual diagnoses of epilepsy and DS. Two patients in this group were not taking any ASMs, 12 were taking 1–3 (average 1.3) of the following ASM: benzodiazepines (CLN, diazepam DZP), VPA, TPM, LTG, LEV, ZNM, GBP, PGB, Carbamazepine (CBZ). All controls had normal EEGs (Appendix A).

The study was approved by the Research Ethics Committee (protocol: IRAS 133481) and conducted in accordance with the Declaration of Helsinki.

### 2.2. EEG

As part of routine hospital recordings in the EMU, scalp EEG was recorded continuously (Natus^®^ Xltek EEG 128 and EEG 46 brainboxes) from 23 channels following the international 10–20 system of electrode placement including Fp1, Fp2, F3, F4, C3, C4, P3, P4, O1, O2, F7, F8, T3, T5, T7, T8, P7, P8, Fz, Cz, Pz, A1, A2 as well as a 2-lead electrocardiogram. Nine patients had additional inferior temporal electrodes as part of the clinical assessment; these were not used in this study. EEGs were sampled at either 256 or 500 Hz. Files were de-identified and exported as European Data Format (EDF) files from the hospital system using Xltek 9.0. Recordings were visually inspected before and after export and re-referencing. Data were read into Matlab using Fieldtrip (ft_preprocessing), down sampled to 256 Hz (if necessary) and re-referenced to mean of the mastoids (A1/A2), except for two recordings with high noise on one mastoid, which were referenced to only the clean mastoid channel.

### 2.3. Sleep Scoring

Sleep was visually scored according to the American Academy of Sleep Medicine (AASM) manual by one expert researcher (JR), aided by clinical annotations to the video EEG that marked sleep stages, seizure times, movement, and awakenings. Sleep was divided into 30-s segments, and each segment was scored. Any 30-s segments of sleep containing movement artefacts or seizures (identified from either clinical annotations or during scoring) were excluded from further analysis. As the amount of total sleep time varied between individuals and groups, the time spent in each stage was normalized to total sleep. The onset of sleep was counted as the first epoch of sleep after 6 pm that led to N2 sleep stage. Any naps prior to 6 pm were excluded from analysis. Three recordings were excluded from analysis due to high electrical interference that prevented accurate scoring, and one was excluded for containing less than one hour of sleep.

### 2.4. Artefact Removal

Cardiac signals detected on the ECGs were subtracted from EEG data using Independent Component Analysis with the fieldtrip script ft_componentanalysis [24]. Subsequently, 30-s epochs were inspected according to variance, inverse variance, and z-scores. Outlier trials were removed using ft_rejectvisual (on average 2.3% of epochs were removed). Channels that showed drift or high noise were interpolated using a weighted average of neighbors using the function ft_channelrepair (<0.5% of channels were removed) [24].

### 2.5. Sleep Spindle and Slow Wave Detection

Following artefact removal, spindles and slow waves were counted across all artifact-free 30-s epochs of stage 2 and 3 on the Cz electrode using custom Matlab^®^ (Mathworks Inc., Sherbom, MA, USA) scripts as previously reported [25]. We selected a broad spindle band of [11–16 Hz] for spindle detection based on a power spectra analysis of 5 to 20 Hz (Figure 2) and subsequent topographical analysis (Appendix A). The root-mean-square (RMS) was then calculated using a 0.2 s time window and putative spindles were identified as events that exceeded 1.5 standard deviations of the individual’s mean RMS. Spindle events were included if they exceeded the detection threshold for between 0.5 and 2.0 s and contained at least 5 complete oscillations and had a local time frequency maxima. Then, a time-frequency transform was applied to putative events and they were marked as spindles only when a unique peak was present in the frequency of the evaluated spindle band. To validate this method of spindle detection in our datasets, we cross-examined spindle detection using three independent techniques. A single 30 min stretch of cleaned data was visually scored by a blinded expert (KH). Then, spindles for all data were automatically detected using both the above algorithm as well as the program SpiSOP (Spindles Slow Oscillation and Power-spectral-density) version 2.3 [26]. The main difference in SpiSOP spindle detection lies in how spindle frequencies are determined. SpiSOP detects spindle frequency bands for each recording by determining peaks in the power spectrum bands automatically, with visual confirmation from the researcher. Frequencies ±1 Hz around the center were defined and bandpass filtered, thus defining a narrower spindle band. Putative events must then exceed 1.5 standard deviations of a smoothed, moving RMS. Spindles must have a duration of 0.5–3 s, exceed a 4 µV threshold at the beginning and end, and not exceed a peak-to-trough value of 200 µV. Both automated detection methods found similar spindle densities (average difference between all datasets of 0.46 spindles/min) and both algorithms showed agreement in the relative spindle numbers. That is, both methods identified the same nights as relatively spindle-rich or spindle-poor, suggesting that both measures were valid to compare spindles across subjects and nights.

Slow waves were detected from the same 30 s artifact-free epochs of stage 2 and 3 that the spindles were detected in as previously reported [25]. Signals were band-passed from [0.5–2 Hz] using a zero-phase FIR filter and events were established as troughs ≤−35.5 μV. Only waves that were bounded by unique zero crossings were counted as slow waves. To remove multi-component artifacts, the duration of the half-waves was challenged, and any subsequent zero crossings that lasted longer or shorter than 0.25–1.0 s was removed.

To compare distributions of spindles (Appendix A), spindle peak-to-trough measurements were rounded to the nearest microvolt and calculated as histograms between 10–90 µV. Data were fit according to a third order gaussian distributions using a non-linear least squares fit (R^2^ > 0.99 for both fits). Difference in distributions determined using a Two-Sample Kolmogorov–Smirnov Test. All values were normalized to the total number of spindles during the night to control for differences in total and stage N2 sleep duration.

### 2.6. Coupling Analysis

The relative coupling strength between slow waves and spindles was determined by first filtering data into slow wave (0.3–4 Hz) range using a first order bandpass Butterworth filter, and determining the instantaneous phase angle of the slow wave band after applying a Hilbert transform. Spindles were filtered using a second order bandpass Butterworth filter from (11–16 Hz). Spindles were detected by utilizing the discrete spindle events, as calculated above. A Hilbert transform was subsequently applied to spindles, and peaks determined as the time where the maximum Hilbert amplitude occurs. For each spindle peak, we tested whether the spindle occurred during a detected legitimate slow wave. Next, one full phase cycle surrounding each spindle was defined for phases between -pi and pi. Any phases outside the detected permitted slow wave duration were excluded from analysis. Subsequently, to smooth the slow wave and prevent noise from affecting the phase calculation, the phase signal cycle was fitted with a 1–5th order polynomial. The fit with the lowest squared error and only one peak (assessed as an absence of a negative slope in the analytic phase) was used. If more than three of the fits included multiple peaks, the event was excluded from analysis (Less than 0.5% of trials were removed with this approach). Surrogate data was also calculated for each of these trials using shuffled indexes [27]. The resulting Coupling strength was determined by using the CircStat toolbox [28] to determine the circular mean of all coupling events in an individual and the strength of the resultant vector, r. Circular means were tested for uniformity using a Rayleigh test and compared against surrogate data. Circular means were additionally bootstrapped with 10,000 iterations to determine confidence intervals of circular distributions.

### 2.7. Statistical Analysis

Statistical analyses were conducted using Matlab R2019 and the open-source toolbox Fieldtrip [24] as well as the open source statistics program Jasp (JASP Team (2020). JASP (Version 0.11.1)) Linear Mixed models were conducted in R (R Development Core Team 2019, version 4.1.3). Equal variances were not assumed for between subject analyses, and all groups underwent two sample F-test for equal variances. Within sample effects of age were examined using linear correlations or treated as a covariance against using Analysis of Covariance (ANCOVA). Unequal variances had Satterthwaite’s approximation applied using the Matlab function ttest2. Variables that were not normally distributed used Wilcoxon tests in place of *t*-tests. In all analyses, significance α levels were set to 0.05. Bootstrap and Montecarlo permutation tests used a resampling rate of 10,000, unless otherwise noted. Circular analyses used the CircStat toolbox [28].

We compared (1) EEG sleep features between nights separately in the Epilepsy group (no ASM to full ASM nights) and Control group (no change in ASM between nights) and (2) the two nights of retrospective sleep polysomnography between groups (Epilepsy and Control), collapsed across nights.

Linear Mixed Models were then used to examine whether individual ASM classes could significantly explain variability in sleep differences. That is, could one or more drug types drive the observed differences or exhibit opposing effects from other ASMs. To prevent overfitting the model to the 13 different medications taken by our sample, ASMs were grouped according to their pharmacological mechanism [29]. The 6 ASM categories were Benzodiazepines (CLB, CLN, DZP with 8 degrees of freedom), Calcium channel inhibitors (GBP, PGB, 7 degrees of freedom), Multimodal mechanisms (TPM, ZNM 8 degrees of freedom), Sodium blockers (LTG, LCM, ESL, CBZ, 15 degrees of freedom). Three drugs with unique mechanisms of action and were not grouped: VPA (10 degrees of freedom), LEV (13 degrees of freedom), PMP (2 degrees of freedom). ASMs were modelled as random effects against the null model, to prevent multiple comparisons. To validate the best model, random effects and interactions between the fixed effects were determined using Repeated Likelihood Ratio Tests and Akaike Information Criteria. Homoscedasticity and normality were determined by visual inspection of residuals. Models were fit with maximum likelihood estimates (ML) using the lme4 package [30] in R and expressed according to the function:SleepOutcome~Diagnosis + RecordingNight + (1|Patient) + (1|ASM).(1)

Significant sleep differences that were found to have a bulk effects of ASM/no ASM as a group were further examined to determine if a single ASM category could explain variance within the model with the presence or absence of ASMs as a main effect and controlling for participant and ASM type according to the function:SleepOutcome~DrugNoDrug + (1|Patient) + (1|ASM).(2)

## 3. Results

### 3.1. Clinical Features

Data from 15 patients (age 19–60; mean 36.4; 8 females, 7 males) diagnosed with focal epilepsy undergoing presurgical evaluation was used for the Epilepsy group and 14 age-matched individuals with DS who acted as the Control group (age 20–65; mean 37.1; 12 females, 2 males) (Appendix A). The majority of patients did not have seizures on either night, with three participants experiencing seizures on the no ASM night and two on the following night with ASMs reinstated. Only one participant had recorded siezures on both nights. Controls had a similar occurrence of dissociative seizures in nights studied, with three patients in the Control group experiencing night siezures and four patients in the Epilepsy group).

### 3.2. Sleep Architecture between Nights and between Groups

Between nights, there was no significant change in total sleep time, time spent in NREM stage 2 (N2), slow wave sleep (N3) or REM sleep (Appendix A, paired, *t*-tests, *p* > 0.05) in either group. The Epilepsy group had slightly more time spent in stage N1 sleep on the first night (no ASM condition) compared to the second night (full ASM), with no change in the Control group (Epilepsy group: *p* = 0.01, median 1.8% more N1 sleep on day 1, Control group: *p* = 0.5).

However, when comparing between the Epilepsy and Control groups, the Epilepsy group showed a significantly shorter duration of N2 (Figure 1B). To understand whether this effect could be due to specific ASMs, we used a Linear Mixed Model to examine the fixed effect of Diagnosis and Recording night on the proportion of N2 Sleep, controlling for participant and validating the model against random effects of each ASM class. Both Benzodiazepine ASMs and Calcium channel modulators significantly explained the variance in night (Benzodiazepines: 37% of variance χ^2^(1) = 6.1, *p* = 0.013; Calcium 19% of variance, χ^2^(1) = 4.0, *p* = 0.046). After correcting for these components, there remained a significant main effect of Epilepsy (t(51) = −2.31, *p* = 0.025), but no significant effect of recording day. No other ASM classes significantly explained the variance, nor did the number of awakenings due to seizures in the night.

The two groups also differed in total sleep time (Figure 1C, Appendix A), with the Control group sleeping longer than the Epilepsy group (total sleep time: Epilepsy group: 5.0 ± 1.1 h, Control group: 6.7 ± 1.7, *p* = 0.00004). To determine if any specific ASMs drove this effect, we fit a linear mixed model of sleep duration with fixed effects of diagnosis and recording night and controlling for participant and each class of ASM. We found that none of the individual ASMs or seizure counts could significantly explain the variance of the data (χ^2^ test *p* > 0.05). and the main effect of reduced sleep in the Epilepsy group remained significant after controlling for age (F(1,49) = 27.8; *p* < 0.001, Figure 1C). One possibility is that the unit protocol can, on an individualized bases, require sleep deprivation in the presurgical evaluation group to increase the likelihood of seizure occurrence on unmedicated nights [31] this practice would not have been carried out on the final night (the night prior to discharge). However, we saw no interaction between recording night and diagnosis in our Linear Mixed Model (t(52) = 0.30, *p* = 0.762 suggesting that clinical sleep deprivation was not a significant factor in the Epilepsy group’s reduced sleep, and we excluded at the outset one participant with less than one hour of sleep.

We then investigated the underlying cause of reduced total sleep time in our sample. We compared whether the groups differed in the time of sleep onset (Figure 1D) or wake (Figure 1E). The Epilepsy group had significantly later mean sleep onset compared to the Control group (Epilepsy group: 1:08 ± 16 min, Control group: 23:27 ± 21 min, *p* = 0.00036) but had no difference in mean wake time (Epilepsy group: 7:15 ± 10 min, Control group: 7:27 ± 18 min, *p* = 0.59), indicating that less sleep in the Epilepsy group came from a later bedtime, accounting for 1.68 of the 1.73 h difference in sleep between groups. Within groups, neither the time of sleep onset nor wake differed according to ASM dose or night (*p* < 0.05, see Appendix A).

### 3.3. Sleep Spindles

First, we performed a very broad range power spectra analysis in the 5 to 20 Hz frequency band to broadly identify changes in the spindle band using a cluster-based permutation of parametric samples from frontal and central electrodes to look for broad changes in spectral power between nights. This analysis showed that in the Epilepsy group between nights with and without ASM, there was a significant effect on power spectra (*p* = 0.030), while no such change in power in this frequency band was observed in the Control group (Figure 2A,B). The significant difference in power spectra in the Epilepsy group was detected in the 10.77–12.00 Hz band, a region within the spindle band, and was higher in medicated nights (Figure 2A). To determine if the power differences reflected spindle activity or non-specific activity in the spindle band, we validated by examining topographic and time-frequency analyses, controlling for benzodiazepines, and examining the effects of fast and slow spindles (Appendix A) showing discrete events consistent with the location expected of fast and slow spindles [32], as opposed to non-specific activity in that frequency band. The increase in power across both fast and slow spindles suggests that ASM can broadly increase power of both types of spindles.

#### Spindle Differences between Night and between Group

As sleep spindle power is derived from three interconnected parameters (amplitude, density, and duration) we next examined whether each of these was affected by ASM treatment. We counted spindles across all artifact-free epochs of sleep stages 2 and 3. Spindle amplitude did not change with ASM (*p* = 0.14, paired *t*-test). A linear correlations of age and spindle amplitude, to control for decreasing amplitude with age and excluding benzodiazepine use, showed a small but significant association between spindle amplitude and group (ANCOVA, F(1,49) =4.06; *p* = 0.049) (Figure 3A). That is, at any given age, the Epilepsy group had higher spindle amplitudes compared to Control group (8% of total variance). Mean amplitudes were 37 ± 1.5 μV for the Epilepsy group and 35.9 ± 2.6 µV for the Control group.

Next, we examined how Inter-Spindle Intervals (ISIs), or the time between adjacent spindles, changed with ASM use. For this analysis, we examined ISIs in 2-s bins from 0–6 s. These intervals were chosen to separate short and long interval ISIs, which may have differential roles in memory reprocessing [33]. We found that within subject, spindles with a 2–4 s interval were specifically reduced with ASM compared to without (*p* = 0.017, paired *t*-test) (Figure 3B). ISIs were not skewed by individuals taking benzodiazepines (*p* = 0.32, unpaired *t*-test) (Figure 3B).

Sleep spindle length for both groups were similar to those reported for the general population (lasting on average for Epilepsy group: 0.82 ± 0.01 s, DS Control group: 0.81 ± 0.01 s, and previously reported for the general population: 0.85 s [34]). Modal spindle lengths had a small but significant reduction with ASM in the Epilepsy group (*p* = 0.039, Figure 3C) with a small to medium effect size (Cohen’s d = 0.35), corresponding to a reduction of about one-half cycle after medication. The Control group showed no change (*p* = 0.49). Spindle density was not significantly altered with ASMs (Control group: *p* = 0.16, Epilepsy group: *p* = 0.28, paired *t*-test) (Figure 3D). Group level effects showed similar trends (Appendix A).

### 3.4. Slow Waves

Using a cluster-based permutation approach, we found a significant decrease in slow wave amplitude with medication (decreased in ASM night compared to no ASM night in the Epilepsy group) (10,000 iterations Cluster alpha threshold 0.05, with two clusters over each slow wave peak, *p* = 0.014, *p* = 0.031) (Figure 4A), corresponding to an ~2.5 µV reduction in peak amplitude with medication (Figure 4A, right). The same approach did not find a significant difference in slow waves between nights in the Control group (Figure 4B). There was no correlation between age and overnight change in slow wave amplitude (Epilepsy: r = <0.001, *p* = 0.99 Control r = 0.075, *p* = 0.34).

Our finding of slow wave reduction and spindle changes with ASM (Figure 2, Figure 3 and Figure 4)were challenged by re-testing each analysis individually in epochs only from stage 2 or 3, to determine if the effects pertained to only one of these stages of sleep. Overall, only the largest effects of ASMs survived subdivision into the two sleep stages (increased spindle power and reduced slow wave amplitude with ASM), both of which retained significance and effect direction across both stages, with decreased slow wave amplitude in ASM night compared to no ASM night, indicating effects were not selective to a specific stage of NREM.

### 3.5. Coupling

Next, we examined whether slow waves and spindles became differentially coupled with medication or between group. Both Epilepsy and Control groups showed tight coupling of sleep spindles to the slow wave (Figure 5B). The strength of coupling (r, the resultant vector length) was significantly reduced by medication (ASM night to no ASM night in the Epilepsy group) (*p* = 0.028, paired *t*-test) (Figure 5B). Coupling was unchanged between nights in the Control group (*p* = 0.61). There was no significant difference between Control and Epilepsy groups in the strength of coupling (Figure 5C).

The mean phase of the slow wave where coupling occurs (θ in Figure 5A) did not differ between group (Epilepsy group: −1.25 ± 2.55°, Control group: −1.58 ± 3.67°, *p* = 0.63, two-way circular-linear ANOVA, F(1) = 0.24) (Figure 5C). The Control group showed almost a 2-fold higher variance in the phase of coupling compared to the Epilepsy group (circular variance Epilepsy group: 0.029, Control group: 0.051). Age significantly correlated with reduced phase of coupling with increasing age (Figure 5C) in both groups (circular-linear correlation, Epilepsy: *p* = 0.003, control: *p* = 0.015). A two-way circular-linear ANOVA, binning ages by decade, found a significant effect of age on coupling (F(4) = 9.9, *p* < 0.001). Although there was no main effect of epilepsy, there was a significant interaction between age and epilepsy (F(4) = 5.9, *p* = 0.0006). That is, the progressive change in coupling during aging differed in the epilepsy cohort compared to controls.

To better understand the driver of coupling strength with ASMs, we correlated other sleep effects that were altered with ASMs but were stable in the Control group. Across individuals in the Epilepsy group, we found a significant linear correlation between the amount that coupling strength changed with medication and the amount that the slow wave amplitude changed with medication (F-statistic *p* = 0.024, R2 = 0.34). In contrast, the control group had no correlation between these values (F-statistic *p* = 0.63, R^2^ = 0.019). (Figure 6A). Similarly, coupling phase was impacted by slow wave amplitude (F-statistic of fit: *p* = 0.047, R^2^ = 0.27), such that as amplitude decreased in the slow wave, the phase of coupling increased, i.e., spindles became coupled closer to the peaks of slow waves (Figure 6B). No such correlation was found in the Control group (F-statistic vs. constant model: 0.58, *p* = 0.46). On the other hand, spindle features impacted by ASM (amplitude and ISI) were independent from coupling changes (amplitude F = 1.1, *p* = 0.32, spindle ISIs in the 2–4 s range (F = 0.014, *p*-value = 0.91)). These findings suggest that ASM-induced changes in slow waves, but not spindles, are related to the reduction in coupling.

To investigate whether any of the specific sleep factors that were broadly affected by ASMs could be significantly explained by a specific drug type, exploratory analysis were conducted using Linear Mixed Models following the format:SleepOutcome~DrugNoDrug + (1|Patient) + (1|ASM)

Models for each finding (spindle amplitude, ISI, spindle length, slow wave amplitude, and coupling) were each modeled. Models were refined by testing if each drug type could improve the model fit. None significantly improved the model fit (χ^2^ test *p* > 0.05), which is unsurprising considering the low sample size for each subtype, although there are interesting trends for Benzodiazepines and VPA towards reducing slow wave amplitude and opposing effects on spindle amplitude. Regressions for each drug type against age can be found in Appendix A.

## 4. Discussion

In this study, we found that comparing sleep from one night of ASM polytherapy to sleep from one night of no ASMs in patients with drug-resistant epilepsy, there were (i) enhanced sleep spindle power, (ii) reduced the number of ISIs of 2–4 s duration, (iii) reduced slow wave amplitude and (iv) reduced coupling between sleep spindles and slow waves, which is known to have an important role in memory consolidation [15,17,35,36]. Both the overall coupling strength and the phase of spindle coupling were linearly correlated with a reduction in slow wave amplitude, in keeping with a close interaction between slow waves and spindles. The opposite did not hold—that is changes in spindle amplitude or timing did not affect coupling. These findings show the effects of ASM on sleep and memory-relevant EEG frequencies in patients with drug-resistant epilepsy. Further, these features may shed light on the previously shown impacts of progressive memory loss in patients with epilepsy [37].

### 4.1. Memory and Sleep in Epilepsy

A fine temporal framing of firing between hippocampal, thalamic and cortical neuronal assemblies takes place during NREM sleep as part of memory reactivation and consolidation, and is necessary to achieve proper information transfer and memory storage [15,17,25]. Three specific waves are generally implicated in this process: predominantly neo-cortical slow waves (0.5–2 Hz), thalamo-cortical fast spindles (11–16 Hz) and hippocampal ripples (80–120 Hz). All three waves are critical for memory consolidation, and their disruption during sleep reduces memory performance [15,16,17,36,38].

Memory impairment is very common in patients with epilepsy [39,40], and is likely due to a combination of many factors including chronic epilepsy pathology [39], seizure burden [40], brain tissue damage [39,40] and the effects of ASMs [41]. It is also likely that the spindle changes we found could indicate poor memory consolidation processes during sleep, as hypnotics that alter spindle density impact verbal memory outcomes [16,42]. Our study found increased spindle power with ASM but did not find a change in spindle density. Previous studies have shown that spindles have a set periodicity of 3–6 s, with an approximate 2–4 s refractory periods, during which memories cannot be replayed [33,43]. In our study, these intervals were lengthened by ASM, with fewer spindles exhibiting intervals in the critical 2–4 s window. This expansion of refractory periods could reduce the total possible amount of replay during a night, reducing memory consolidation, or enact morphological differences in replay efficacy.

### 4.2. Mechanisms of Memory Consolidation Potentially Impacted by ASMs

Emerging evidence has suggested that memory consolidation relies on a tight temporal relationship between ripples, spindles and slow waves [15]. This relationship is maintained by synchronized changes in membrane polarization, where the rhythmic slow hyperpolarization and depolarizations of the neuronal membrane potential during slow waves open windows of excitability for thalamic, cortical and hippocampal networks [10,15,44,45,46]. The spindles then activate ripples, acting in a top-down approach [17,35,47,48]. This process is further entrained through bottom-up ripple and spindle activity [35,46]. The excitability and passive cooperative dynamics of the system are susceptible to decoupling with age or reduced grey matter [36,47]. Slow- wave-spindle coupling in our study was reduced with age, as previously shown in healthy controls [47]. However, longitudinal studies of patients with uncontrolled epilepsy report a strong progressive memory decline [39]. In line with this, our findings show that decoupling of spindles and slow waves with age interacted differently in epilepsy compared to controls. Since this study did not test memory or memory consolidation, future studies are needed to determine whether these slow wave decoupling events are sufficient to impact memory consolidation.

In this study, we found that ASM polytherapy reduced sleep slow wave amplitude and this reduction was linearly correlated with reduced sleep spindle/slow wave coupling strength and a shift towards a later coupling at the peak of the slow wave. This is consistent with an impaired top-down coupling system [15,45,46]. In healthy individuals, spindles are phase-locked to the negative-to-positive transition of the slow wave, representing cortical neurons transitioning from hyperpolarized to depolarized states [15,17,47]. Slow wave amplitude is also reduced in other situations, such as old age or reduced grey matter. In these cases, the time where slow waves and spindles are coupled is delayed until the cortical neurons are depolarized ie at the peak of the slow wave [35,47]. The reverse effect is seen in with slow waves and high frequency oscillations in epileptic zones, where high amplitude promote an earlier coupling phase [13,49]. Our finding are consistent with the literature in other conditions to show that decreases in slow wave amplitude are associated with a direct and linear reduction in coupling.

One unique difference in the mechanisms of ASM decoupling, compared to age or grey matter loss, is that while slow waves were reduced, spindle power was actually enhanced, while aging reduces both waveforms. Spindles have been shown to reinforce the circuit in both top-down and bottom-up approaches [16,46]. Our findings suggest that, in this case, the enhanced spindle power was not sufficient to recover the slow wave-spindle coupling associated with the slow wave amplitude were reduction.

### 4.3. Limitations

There are number of limitations to our study. This study was a retrospective analysis of sleep data, and as such, we could not ultimately control for, or understand, factors that were not recorded. Additionally, we could not measure if changes in memory-relevant circuitry translated to memory impairments in the period of the study. Further studies are needed to explore memory consolidation and the effects of ASM treatment. Additionally, our cohort was heterogenous and the variability in age and ASM and epilepsy type could have masked both disease and drug specific effects that could be visible in larger or more homogenous datasets. Our control group was selected as a control for the effect of sleeping in the hospital environment and comparisons between 2 nights, were age-matched patients and had no change in their medication intake. However, controls could not be sex matched or drug (ASM) matched.

This dataset was developed to examine sleep impairment experienced by real-world patients with multiple interacting medications. In order to model individual ASM mechanisms driving sleep impairments, we used Linear Mixed models to examine effects of each medication as a random effect. However, this does not control for interactions between drugs, and subgroup analysis was limited by relatively small sample sizes and larger studies are needed to understand the roles of individual ASMs. Additionally, some ASMs (VPA, LTG, ZNM, CLN) have a half-life that may exceed 24 h and could therefore still be at active levels in some patients in the no-ASM condition (blood concentration half lives determined using medication repositories (FDA.gov/search and https://www.medicines.org.uk/emc, accessed on 29 March 2022)). However, decay calculations from their 5-day records show that all patients had at minimum a 4-fold lower concentration of ASMs compared to pre-admission baselines.

As such, interpretations should be handled cautiously and should not be applied to all patients with epilepsy.

We found that total sleep time was shorter in the epilepsy group due to a later sleep onset. In contrast, two previous studies that specifically examined sleep architecture in DS compared to epilepsy in a hospital environment found no difference in total sleep time, and in fact, one found later bedtimes in the DS group, which may reflect the variability within the patient groups or the challenges inherent in scoring fragmented sleep commonly co-occurring in people with epilepsy [50,51,52,53]. However, a recent meta-analysis that combined 24 studies was able to show that sleep efficiency, which is a measure of time spent awake during the night, is significantly reduced in focal epilepsy compared to healthy controls, and persists regardless of ASM treatment [29]. Although our study does not differentiate insomnia from chosen later bedtimes, it suggests that reduced sleep efficiency in Epilepsy could arise from later sleep onset, rather than from purely night time awakenings.

## 5. Conclusions

In conclusion, this study found that ASM polytherapy had significant impacts on NREM sleep waveforms in patients with drug-resistant epilepsy. This study found that, as a group, ASM polytherapy did not cause a general shift in time spent in any particular sleep stage, but that within the groups, patients taking Benzodiazepines and Calcium channel modulators had abnormal amounts of N2. Further, ASM did not have a blanket effect on silencing waveform amplitude, but rather a specific decoupling of slow waves and spindles correlated with slow wave amplitude and interacting with age. Future studies are needed to determine the effects of these ASM related sleep modifications on memory consolidation in focal epilepsy.

## Figures and Tables

**Figure 1 brainsci-12-01288-f001:**
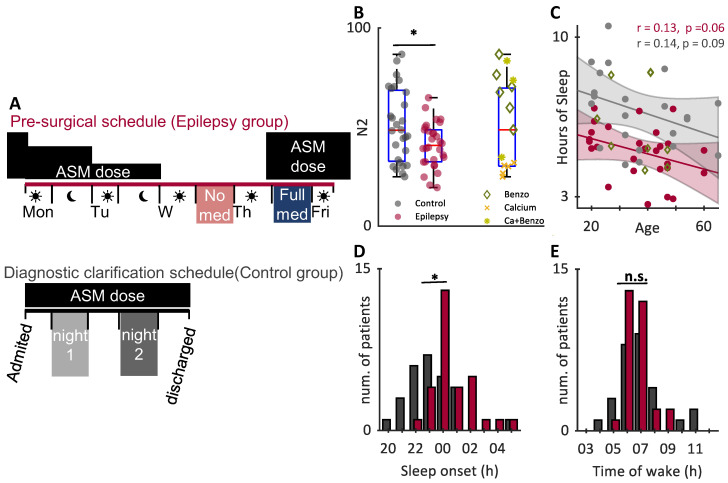
Differences in sleep between groups. (**A**) Cartoon showing the schedule for both the Epilepsy and Control groups. (**B**) Normalized time spent in N2 sleep between groups significantly differed between the Epilepsy and Control groups (left bars, unpaired *t*-test with unequal variances, *p* = 0.019). High variance was caused by individuals that were taking benzodiazepine (green diamonds) or Calcium modulators (yellow X) or both (light green stars). Colour code in (**B**) applies to (**C**–**E**). (**C**) Linear correlation of the age of participants vs. total hours slept. Points represent a single patient’s night of sleep; bands show the 95% confidence interval of linear fit. Sleep differences between groups remained significant after controlling for age and excluding benzodiazepine use (ANCOVA, Age F(1,49) = 5.9; *p* = 0.019, Diagnosis F(1,49) = 27.8; *p* < 0.001). (**D**) Histograms of sleep onset times and € wake, according to a 24 h clock. Sleep onset times were significantly later in the Epilepsy group (red) compared to Control group (grey; mean sleep onset: 1:08 for epilepsy, 23:27 for control). Wake times did not differ between group (unpaired *t*-test, *p* = 0.59) and neither sleep onset nor wake differed within group (*p* > 0.05). * *p* < 0.05, n.s. *p* > 0.05.

**Figure 2 brainsci-12-01288-f002:**
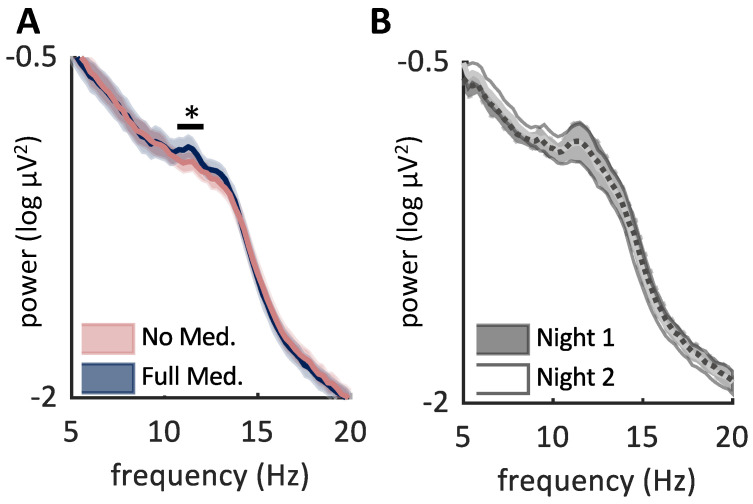
Spindle power is enhanced with anti-seizure medication. (**A**) EEG power in the 5–20 Hz frequency band for stage N2 and N3 sleep in the Epilepsy group (left) and Control group (right). Cluster based permutation test across frontal and central electrodes revealed a significant difference between the night where ASM were given (blue) compared to no medication (pink) in patients with epilepsy (*p* = 0.030, black band shows the significant range from 10.77 to 12.00 Hz). Graph shows mean ± SEM. (**B**) No difference was found between two sequential nights of sleep in the Control group (10,000 iterations *p* < 0.05). Graph shows mean ± SEM. * *p* < 0.05.

**Figure 3 brainsci-12-01288-f003:**
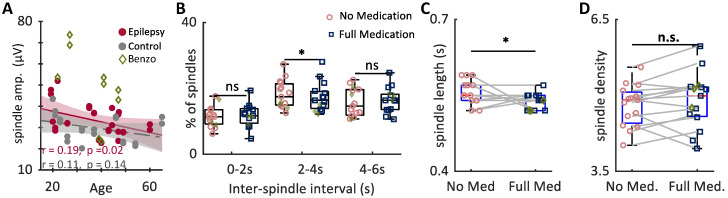
Spindle amplitude is higher in patients with epilepsy and spindle length and the amount of 2–4 s inter-spindle intervals are reduced with anti-seizure medications. (**A**) Linear regression of spindle amplitude controlling for age and benzodiazepine use. The Epilepsy group (red) shows significantly higher amplitudes across ages (ANCOVA, F (1, 49) = 4.06; *p* = 0.049, excluding benzodiazepine users (green diamonds). (**B**) The time between spindles were reduced with ASM at the individual level and were specific to a reduction in spindles with a 2–4 s inter-spindle interval (*p* = 0.017). Changes in inter-spindle interval were not driven by benzodiazepine treatment, as benzodiazepine users did not differ from non-users (unpaired *t*-test, *p* = 0.32, green diamonds). (**C**) Individuals show a small but significant reduction in modal spindle length with medication (paired *t*-test, *p* = 0.039). (**D**) Spindle densities did not differ with medication (*p* = 0.28, paired *t*-test). * *p* < 0.05, n.s. *p* > 0.05.

**Figure 4 brainsci-12-01288-f004:**
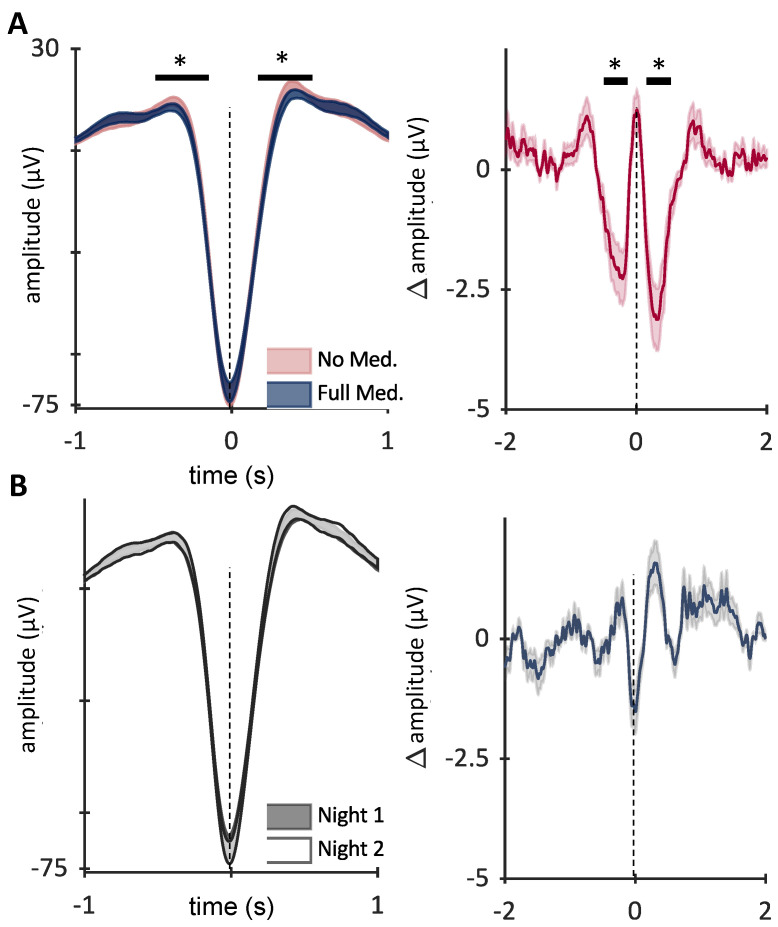
Slow wave peaks are reduced with anti-seizure medication. (**A**) Trough-locked Slow waves on nights without medication (red) compared to those with medication (blue). Slow wave peaks were lower on the ASM treatment night (blue, cluster permutation 10,000 iterations Cluster alpha 0.05, *p* = 0.014, *p* = 0.031, parametric dependent samples). On the right, the average difference between night for each patient (slow wave amplitude with ASM- no ASM) is quantified. That is, ASM use reduced slow wave peaks ~2.5 μV. (**B**) No significant difference was found between two control nights. Graph shows mean ± SEM. * *p* < 0.05.

**Figure 5 brainsci-12-01288-f005:**
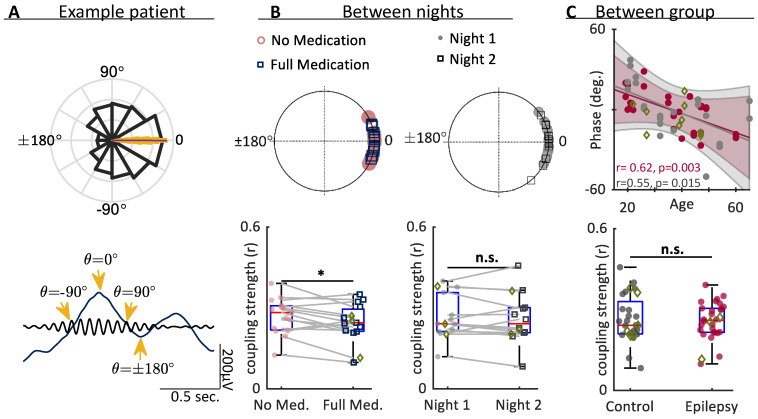
ASM medication reduces spindle-slow wave coupling. (**A**) Example data from a patient in the Epilepsy group. Upper panel is a normalized histogram of the slow wave phases that coincided with spindle peaks (black). Data were non-normally distributed (*p* < 0.001, z = 65.6, Rayleigh test, resultant vector length r = 0.22). Overlaid yellow bar shows the bootstrapped mean (10,000 iterations, *p* < 0.001, Watson-Williams parametric test of equal means). Lower plot shows a single filtered spindle (black) and slow wave (blue) with the phase of the slow wave labelled (orange arrows). (**B**) (upper circular plots) Each point represents the mean coupling phase for a single night of data, with values calculated as in part A. Both groups show tight coupling to the peak of the slow wave. Nights without medication are in pink, nights with medication are in blue. (lower plots) Coupling strength, as determined by the vector length, r, was significantly reduced with ASMs (*p* = 0.028, paired *t*-test), but was unaffected between nights in the Control group (*p* = 0.61). (**C**) The phase of coupling correlated with age in both epilepsy (*p* = 0.003, correlation coefficient 0.62) and control (*p* = 0.015, correlation coefficient = 0.55) groups (circular-linear correlation). A two-way circular-linear ANOVA, binning ages by decade, found that age had a significant effect on coupling phase (F(4) = 9.9, *p* < 0.001), but there was no main effect of diagnosis (F(1) = 0.24, *p* = 0.63), but diagnosis did interact with age (F(4) = 5.9, *p* = 0.0006) (below). The two groups did not differ in the strength of phase coupling (on either night, *p* > 0.05, unpaired *t*-test). Epilepsy group shown in pink, controls shown in grey, * *p* < 0.05, n.s. *p* > 0.05. Green diamonds show those taking Benzodiazepines.

**Figure 6 brainsci-12-01288-f006:**
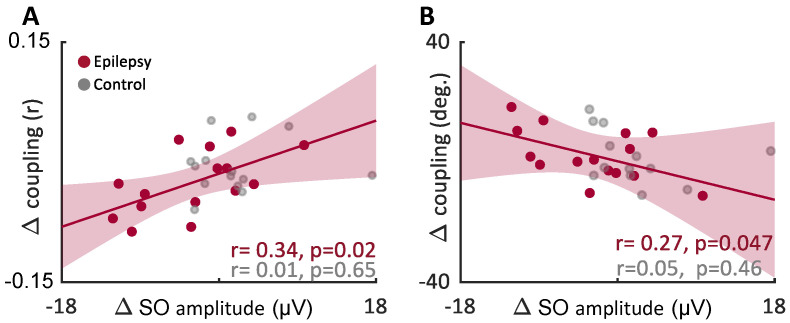
Medication-related changes in coupling and slow wave amplitude are correlated: (**A**) Linear correlation of the change in coupling strength and the change in Slow Wave (SO) amplitude after ASM polytherapy (red), or between nights in the Control group (grey). Each point represents the change between nights for an individual. A reduction in slow wave amplitude is correlated with a significant reduction in coupling strength (F-statistic of fit *p* = 0.024, R^2^ = 0.34). The Control group had no such correlation (F-statistic *p* = 0.63, R^2^ = 0.019). (**B**) The phase of coupling was similarly correlated, such that a reduction in slow wave amplitude was correlated with an increase in phase, shifting the phase closer or even past the slow wave peak of 0° (more depolarized) (F-statistic of fit: *p* = 0.047, R^2^ = 0.27). The Control group showed no such correlation across day (F-statistic of fit *p* = 0.46, R^2^ = 0.04).

## Data Availability

Data available on request due to restrictions (eg privacy or ethical). The data presented in this study are available on request from the corresponding author. The data are not publicly available as there is no consent to do so but anonymized data can be made available to bona fide requests.

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
