# Peer review of "Effects of Anti-Seizure Medication on Sleep Spindles and Slow Waves in Drug-Resistant Epilepsy"

_brainsci, 2022, doi:10.3390/brainsci12101288_

Round 1

Reviewer 1 Report

In the table S1 lack describes the abreviature DS

Statistic analysis

In the statistical analysis they did not include the effect of age, a factor that could help explain the uncoupling mechanisms of the ASM. In addition, in the discussion they could include the age ranges that slow wave sleep is reducing, since the authors have age ranges of 20-30, 30-40, 40-50 and subjects older than 50 years of age.

Discussion

The authors could improve the discussion regarding the results obtained in sleep architecture. I suggest reviewing the following article. Unlike the present study, the control group are subjects with DS with ASM, in the review they compared the sleep architecture was compared between patients with epilepsy and healthy controls.

Wei-Chih Yeh, Huan-Jan Lin, Ying-Sheng Li,    Ching-Fang Chien , Meng-Ni Wu , Li-Min Liou , Cheng-Fang Hsieh, Chung-Yao Hsu , Rapid eye movement sleep reduction in patients with epilepsy: A systematic review and meta-analysis. 2022 Mar;96:46-58.

 doi: 10.1016/j.seizure.2022.01.014. Epub 2022 Jan 29.

It is suggested to include the effects of ASM on the memory process

At line 545-550. The idea is not clear about the difference in the mechanisms of uncoupling of ASM, compared to age or loss of gray matter.

 In the line 566-568 I suggest considering the half-life of ASM, a variable that could influence the results. include references

Reviewer 2 Report

The paper of J.K. Roebber et al. describes the influence of various anti-seizure drugs on sleep and phasic sleep EEG events. The authors used two cohorts of patients: those with dissociative seizures (n=14, Control group) and those with drug-resistant epilepsy (n=15, Epilepsy group, including 10 patients with temporal lobe epilepsy). In my opinion, this paper needs to be revised for clarity. Please find my critical remarks below.

What were the main goals of this research (in Introduction)? Considering that this study focused on "Effects of anti-seizure medication (title)", why were the effect of epilepsy and the effect of age examined?

The study design is a bit complicated. I did not understand the reason for selecting patients with DS as control, considering that 2 out of 14 patients did not take any drugs; 8 patients were treated with monotherapy and 3 - with polytherapy. In the group of patients with epilepsy, some patients were deprived of sleep (Line 297). The factor of sleep deprivation could significantly affect the outimes.

The authors selected two spindle types, but did not explain why they were interested in anterior and posterior sleep spindles. There are many reports about anterior and posterior spindles in humans. Please clarify the issue of selecting these two spindle types in the present report.

How medication influenced two spindle types, considering that some effect was defined in 10.77-12.00 Hz?

Line 173. Spindles were detected using "the program SpiSOP (Spindles Slow Oscillation and Power-spectral-density) version 2.3 [26]." The reference [26] seems incomplete.

The number of sleep spindles was measured, and spindle density was computed. What epochs were used for analysis? Sleep Stages 2 and 3? How often sleep spindles coincided with slow waves? Did the authors differentiate between single spindles and spindle-slow-wave complexes?

Line 239. "Linear Mixed Models were used to examine whether individual ASM classes could significantly explain variability in sleep differences." This model seems to be a key method used in the present study. Could the authors describe this model in details?

Line 331-332 "A significant cluster was detected in the 10.77-12.00 Hz band, a region within the spindle band, showing higher power in medicated nights." Unclear to me.

Figure 1.B demonstrates a mixture of data obtained from the group with epilepsy and DS (control group) with different types of pharmacotherapy. I am confused by this presentation of results.

Figure 1.C demonstrates age-related differences in sleep duration (i.e., hours of sleep). The significance of the factor 'AGE' has to be evaluated statistically.

The effect of benzodiazepine treatment on sleep and sleep spindles is interesting, but it was predictable (it is well known from literature). Age-related changes in sleep architecture and sleep EEG have also been well studied. Did the authors have hypotheses about the age-factor and benzodiazepine-factor?

Line 394-399. Unclear.

Figure 4.A. I do not see significant difference between the parts marked with asterisks. In Fig.4, both right parts are unclear to me. What is DeltaAmp shown as y-axis? Results (line 187): "same artifact-free epochs of stage 2 and 3 that the spindles were detected in".... How long was this epoch? How strong was the effect of anti-seizure medication? How strong was the effect of epilepsy? How strong was the effect of age?

Section 3.4. Coupling. How many samples were analyzed?

Line 452. "reduction in coupling strength" Reduction. What was the baseline?

Line 454-455. "indicating that changes in slow waves may underlie the reduction in coupling (Figure 6A)" In my opinion, this conclusion is not supported by the results presented.

Line 463-464. "Linear Mixed models for spindle amplitude, ISI, spindle length, slow wave amplitude, and coupling were individually fit against a bulk drug effect controlling for participant." It seems to be a very complex model. Please clarify this issue.

Conclusion could be more specific and precise. Line 580: "individual ASM have been shown to affect sleep staging" What were individual ASM and how were they affected sleep?

The authors mentioned "memory consolidation" in Introduction and Discussion, but do not present data on this point.

Minor.

Line 10. "... sleep and epilepsy where each worsens the other". Sleep worsens epilepsy and vice versa? There are many types of epilepsies, and only some of them are associated with sleep and sleep disorders.

Line 299. "nights30this"

Line 261-263 "This section may be divided by subheadings. It should provide a concise and precise description of the experimental results, their interpretation, as well as the experimental conclusions that can be drawn." Please remove these lines and give a concise and precise description of the experimental results.

Line 368. 0.85. seconds29).

Line 376. Should be Figure 3

Reviewer 3 Report

This paper entitled “Effects of anti-seizure medication on sleep spindles and slow waves in drug-resistant epilepsy” described a study of the relationship between anti-seizure medication (ASM) and sleep. They compared the effect of ASM treatment on sleep and sleep in patients who were drug-resistant during pre-surgical evaluation. They found increased spindle power and decreased slow wave amplitude in ASM-treated patients.

Overall, this paper employed a lot of mathematic models and apply it to clinic data, there are some novel points in the methodology section. However, a few points of the conclusion are not supported by the data presented in the current paper. Moreover, this manuscript contains a lot of typo, spelling and sentence structure mistakes that lead to confusion and misleading to readers.  

Major concerns:

1.     The data analysis in this paper is based on a small sample collection, more samples are necessary to make the conclusion solid. For example, there are only two males in the control group, will the unbalanced sex distribution affect the conclusions.

2.     In “conclusion” section: the authors claimed that “ASM did not cause a general shift in time spent in any sleep stage”, does it conflict to the results that “sleep onset times were significantly later in epilepsy group compared to the control” (Figure 1D)?

3.     In “discussion” section, the authors spent a lot of space to discuss the relationship between memory consolidation and sleep. Although memory loss is a common phenomenon is patients with epilepsy, but there is no solid evidence to support the link between memory consolidation and slow waves etc. Additional experiments (memory test) are needed to prove that ASM can cause memory impairment.

Minor concerns:

1.     Line 151-152: need reference.

2.     Line 218-219: need reference.

3.      Line261-263: please delete the instruction.

4.     Line: 272: should be ”between nights”.

5.     Line 280: “,” missing.

6.     Line 299: should be “[30]”.

7.     Line302: delete “a”.

8.     Line 309: change ”lower” to “less”.

9.     Line310:  delete space between “h” and “o”

10.  Line 326: explain why only use 5-20HZ for sleep spindle analysis.

11.  Line342: there is a black line in figure 2A which is not annoted.

12.  Line 353: should be”ANOVA”.

13.  Line 360: “[31]”.

14.  Line368: “[29]”.

15.  Line 376: should be Figure”3”.

16.  Line389: from Figure 4A, it’s increased amplitude in ASM (red).

17.  There are more format issues in the reference list, more carefully proofread is needed.

Round 2

Reviewer 2 Report

All my questions were answered.

Reviewer 3 Report

I'm pleased to see that the authors addressed all my comments and questions and don't have any further questions prior to its publication.